# Facile Fabrication of Flexible Polymeric Membranes with Micro and Nano Apertures over Large Areas

**DOI:** 10.3390/polym14194228

**Published:** 2022-10-09

**Authors:** Kebin Li, Javier Alejandro Hernández-Castro, Keith Morton, Teodor Veres

**Affiliations:** 1National Research Council of Canada, 75, de Mortagne, Boucherville, QC J4B 6Y4, Canada; 2Currently is working with Symgery, 224 Rue de l’Hôpital, Montréal, QC H2Y 1V8, Canada

**Keywords:** polymeric membrane, micro- and nanostructured membrane, UV polymerisation, spontaneous capillary flow, hot embossing, polyvinyl alcohol (PVA), polylactic acid (PLA), micro-nano photo-lithography process

## Abstract

Freestanding, flexible and open through-hole polymeric micro- and nanostructured membranes were successfully fabricated over large areas (>16 cm^2^) via solvent removal of sacrificial scaffolds filled with polymer resin by spontaneous capillary flow. Most of the polymeric membranes were obtained through a rapid UV curing processes via cationic or free radical UV polymerisation. Free standing microstructured membranes were fabricated across a range of curable polymer materials, including: EBECRYL3708 (radical UV polymerisation), CUVR1534 (cationic UV polymerisation) UV lacquer, fluorinated perfluoropolyether urethane methacrylate UV resin (MD700), optical adhesive UV resin with high refractive index (NOA84) and medical adhesive UV resin (1161-M). The present method was also extended to make a thermal set polydimethylsiloxane (PDMS) membranes. The pore sizes for the as-fabricated membranes ranged from 100 µm down to 200 nm and membrane thickness could be varied from 100 µm down to 10 µm. Aspect ratios as high as 16.7 were achieved for the 100 µm thick membranes for pore diameters of approximately 6 µm. Wide-area and uniform, open through-hole 30 µm thick membranes with 15 µm pore size were fabricated over 44 × 44 mm^2^ areas. As an application example, arrays of Au nanodots and Pd nanodots, as small as 130 nm, were deposited on Si substrates using a nanoaperture polymer through-hole membrane as a stencil.

## 1. Introduction

Micro- and nano-porous membranes have a wide range of applications, including plasmonics, data storage, energy devices, as well as filters and separators, for biomedical applications. Nanoporous membranes with very well defined pore sizes are based on silicon, silicon nitride, or highly periodic anodic alumina substrates and are typically made using microfabrication techniques. For example, suspended silicon nitride (SiN) membranes were used directly as shadow masks to pattern infrared plasmonic nano antenna arrays [1], sub-50 nm nanoparticles [2] and metallic nanowires [3]; ultrathin and highly periodic anodic alumina membranes were used as a lift-off masks to deposit ferroelectric nanocapacitor arrays for data-storage applications [4] or as templates for direct synthesis of single-crystalline nanopillar arrays in solar cells [5]; SiN membranes coated with biomimetic molecules to form nuclear pore complexes were applied to study nucleocytoplasmic transport phenomena at the single-molecule level in vitro [6]; ultrathin pnc-Si membranes were used for macromolecules separation [7]; Si, SiN, and SiO_2_ nano membranes are now frequently used to measure the translocation of individual DNA molecules [8]; a Si-based membrane with extremely small nano pores fabricated via a self-assembled monolayer deposition of 3-mercaptopropyltrimethoxysilane was used for inorganic–organic proton exchange in fuel cells application [9]. These Si-based membranes are mechanically stable and offer advantages by maintaining the membrane’s shape during handling, but they are nevertheless both fragile and brittle. Furthermore, the microfabrication process for Si-based membranes usually involves specialized, complicated and expensive equipment in dedicated cleanroom facilities.

Alternatively, flexible polymeric membranes are less expensive to fabricate and offer numerous advantages, such as conformal wetting and easy peel-off without significant damage and distortion. Polymeric membranes are also increasingly attractive for biological applications. For example, flexible membranes made of polydimethylsiloxane (PDMS), polystyrene, polyethylene glycol diacrylate, polycarbonate (PC) or thermoplastic elastomer (TPE) have been integrated into bio-inspired microfluidic devices for drug screening and toxicology applications [10], for efficient culture and analysis of renal tubular cells [11], for isolation of rare circulating tumor cells [12] and also as key components of 3D microfluidic systems [13,14,15,16,17,18]. However, it is non-trivial to make polymeric membranes with regular, straight and open-through pores because it is quite challenging to obtain ‘free standing’ and ‘residual-layer-free’ structures in the fabrication of polymeric membranes, particularly for smaller pore sizes. It is, therefore, desirable to make low-cost polymeric membranes with regular micro- and nano-pores to replace the expensive and fragile Si-based membranes.

Various techniques to make polymeric membranes were previously reported. Track- etched polycarbonate (PC) membranes with pore sizes ranging from 100 nm up to 20 µm are readily available commercially. However, the pores in PC membranes are irregular and discrete. Through holes in track-etched membranes are not usually straight because they are formed through a combination of charged particle bombardment (irradiation), followed by chemical etching. Previously, regular, straight open-through-holes TPE membranes were successfully fabricated by using the hot-embossing methods [18]. Although this method could be applied to fabricate PMMA open-through-hole membranes with both high aspect ratio and sub-micrometer diameter pores, the method remains hard to scale-up due to low throughput and the need for expensive metallic molds [19]. An alternate fabrication route for regular, straight, open-through-hole membranes in PDMS elastomers used advanced techniques, such as the micromolding in the capillaries (MIMIC) method [20]. There are several limitations in this technique, including the intrinsic low aspect ratio of pores in PDMS, making it difficult to fabricate sub-10 µm pore sizes. Micro-sieves made of polyimide or polyethersulfone were also successfully fabricated by using phase separation micromolding methods [21,22]. Very recently, Cho et al. [23] demonstrated the fabrication of UV-cured fluorinated perfluoropolyether (PFPE) polymer membranes by a hierarchical mold-based dewetting method, similar to the MIMIC method above. Their fabrication strategy for PFPE nano-apertures leverages the large wettability difference during micromolding and the geometric reinforcement of a multiscale, multilevel architecture. The technique is, therefore, highly material dependent. The mechanical peel off technique that was applied to release membranes also risks membrane rupture, particularly for thin membranes over very wide surface areas.

Previously we proposed a fabrication strategy for freestanding, high porosity, large-area and defect-free micro- and nanoporous polymeric membranes using vacuum-assisted UV micro-molding (VAUM). We demonstrated their use for particle and cell enrichment, including the capture and release of white blood cells and circulating tumor cells [24,25,26]. Here, we present a simplified alternative to this method using spontaneous capillary flow (SCF), which offers a robust pathway to reduce fabrication costs and scale membrane manufacturing. In the SCF method, the final polymer membrane is replicated from an intermediate, sacrificial template. Both UV curing and thermal setting were used for final membrane polymerisation. Membranes were separated from the sacrificial templates by submerging in aqueous solutions for a short time; alleviating the need for mechanical demolding reduces risks of permanently distorting or breaking the membrane. We used both polyvinyl alcohol (PVA) and biodegradable polylactic acid (PLA)/thermoplastic starch blends [27,28] as sacrificial templates. The intermediate templates were fabricated by hot-embossing PLA sheets extruded from PLA pellets. PVA is a highly hydrophilic polymer with good thermal stability, high hydrolysis and the ability to dissolve readily in water with negligible effects on the final polymeric membrane. For PVA intermediate templates, membrane separation is simply carried out in a water bath. For intermediate templates made from PLA, membrane separation was done in aqueous conditions using Nano-Strip 2X^®^ (Cyantek KMG 539400, Amplex Chemical Products Ltd., Pointe-Claire, QC, Canada). We note that for certain classes of polymers that have a tendency to swell reversibly, an intermediate template could also be made by using cyclic olefin copolymer (COC-Zeonor 1060R, Zeon Chemicals, Louisville, KY, USA), which can be fabricated rapidly and cost effectively by hot embossing, and polar solvents, such as methanol, can be used to separate the membrane from the template.

## 2. Experiment

### 2.1. Process Flow Chart for Fabrication of Polymeric Membranes

Figure 1 shows the general process flow chart for the fabrication of polymeric membranes using an intermediate template. The space between micro/nano pillars is filled by polymeric resin via spontaneous capillary flow. Polymerisation is then initiated by UV curing or thermal setting. Separation of the cured membrane from the template is performed under liquid conditions. For most cases presented here, the template is made of PVA and is simply dissolved by deionized water. For PLA templates, Nano-Strip 2X^®^ was used to dissolve the intermediate template and release the polymeric membrane. Alternatively, separation of polymeric membranes could be done using polar solvent, such as methanol, if polymer swelling is an issue.

### 2.2. Fabrication of Silicon Mold Masters and PVA, PLA Intermediate Templates

A typical photolithography process for making a Si master mold is summarized as follows: A single side polished Si wafer (150 mm in diameter, 650 µm in thickness, resistivity of 1–100 Ω·cm and in orientation of <100>) was first surface treated with O_2_ plasma (Oxford PlasmaLab 80 Plus, 200 W, 100 mTorr, 2 min). AZ5214E photo resist (Microchemicals GmbH, Ulm, Germany) was spin coated on the clean Si wafer (2000 rpm, ramp rate of 400 rpm/s, 35 s) followed by a soft bake at 120 °C for 60 s on a hot plate. It was then exposed by i-line UV photolithography at 120 mJ/cm^2^ (EVG^®^ 6200 NT). The photoresist was developed in AZ400K (1:3 = AZ400K:DI water). Any residual photoresist was then descummed by O_2_ plasma (100 W, 100 mTorr, 30s). The wafer was hard baked for 2 min at 110 °C before deep reactive ion etching (DRIE). A Bosch process was used in the Silicon DRIE with alternative 7 s Bosch deposition (with HF power of 10 W, ICP power of 700 W, total gas pressure of 30 mTorr (gas flow rate of 1 sccm for SF_6_ and 50 sccm for C_4_F_8_)) followed by 7 s Bosch etch (with HF power of 25 W, ICP power of 700 W, total gas pressure of 30 mTorr (gas flow rate of 1 sccm for C_4_F_8_ and 50 sccm for SF_6_)) for 92 cycles (PlasmaLab System 100, Oxford Instrument, Yatton, UK). After DRIE, the Si wafer was cleaned with Acetone and O_2_ plasma cleaning for 7 min with 200 W RF power and O_2_ pressure of 100 mTorr. The typical target etch depth of the patterned Si structures was 30 µm. In order to fabricate a Si master mold with 100 µm tall micropillars, both the etching time and the number of Bosch process cycles were increased; a thicker AZ9260 photoresist was used in this case.

A PDMS replica was then created from the etched silicon master (pillars) by pouring a mixture of SYLGARD 184 Silicone elastomer base and its curing agent (Dow Corporation, Midland, MI, USA) in 10:1 by weight over the Si master mold. After curing at 80 °C for 2 h in an oven, the PDMS replica (holes) was demolded from the Si master. In order to facilitate the separation of the PVA intermediate template from this PDMS replica, the surface of the PDMS mold was coated with a monolayer of trichlorol(1H, 1H, 2H, 2H)-perfluorooctyl-silane (97%) (Sigma-Aldrich, Oakville, ON, USA) by placing it under vacuum in a desiccator for two hours. The intermediate template was made by replicating the structures in polyvinyl alcohol (PVA, Sigma-Aldrich) from the PDMS mold. A liquid PVA solution was made by dissolving 20 wt. % of PVA, which has a hydrolysis degree of 89% and a molecular weight of about 100,000 g mol^−1^, in water. The PVA solution was poured over the PDMS mold and placed under vacuum for one hour to remove any trapped air. The PVA was then solidified by slow drying in an oven at 50 °C for 12 h. To ease handling, the preferred thickness of the PVA templates is 300 µm or more. The replicated PVA template could then readily detach from the PDMS mold without any sticking issues.

Replication of PVA intermediate templates from Si master molds using PDMS replica becomes more difficult when the micropillar diameter is less than 5 µm. This is due to the commensurate increase in pillar density. For membranes with smaller pore size, we made intermediate templates poly lactic acid (PLA). PLA is a thermoplastic, biodegradable polymer derived from renewable bio-resources, such as corn, sugar beet and potato starch. The PLA/thermoplastic starch blends are becoming more attractive because of their promising developments for applications in food packaging and biomedicine [27]. PLA or PLA/TPS can be extruded from PLA pellets or TPS, PLA pellet mixtures [29]. Hot embossing can also be used to fabricate PLA-based microstructures [30]. Here, we demonstrate hot embossing of 2 µm diameter micropillars in PLA with an array density of more than 40%. We used a flexible fluorinated ethylene propylene (FEP) working mold to facilitate the demolding process of the PLA intermediate template. Again, a Si master mold (pillars) was first fabricated using standard photolithography and DRIE etching. The FEP replica mold (holes) was created from the etched Si master by hot embossing (EVG^®^ 520HE, EV Group, Schärding, Austria, 10 kN, 280 °C, 30 min). The FEP was then used to emboss PLA substrates (15 kN, 180 °C, 10 min) to create the PLA intermediate template (pillars).

In order to make polymer membranes with nano-apertures, it is very useful to apply a strategy of geometric reinforcement into the membrane fabrication process via multilevel, hierarchical architectures. To do this, we designed a master mold with a combination of nano- and microstructures. Here, the silicon master mold was fabricated by both e-beam lithography and photo lithography and etching. An array of 300 nm square, 600 nm tall nanopillars (10 × 10 mm^2^ patterned area) were fabricated by e-beam lithography [31] and arranged in centered hexagonal configuration the distance between the nearest neighbors fixed at 600 nm. Photolithography and DRIE etching were then used to pattern an array of 15 µm diameter, 30 µm tall micropillars (40 × 40 mm^2^ patterned area).

Although PDMS can replicate the inverted sub-micrometer features from a master mold, it is extremely challenging to replicate either PVA or PLA intermediate templates with hybrid micro-nanostructures from this PDMS. Alternatively, the nanostructures on the silicon master mold are well-replicated by hot embossing into thermoplastic substrates using a secondary intermediate replica molding step in UV-cured polymer (Solvay MD700). The use here of fluorinated polymer working stamp facilitates the demolding process after hot embossing. The working stamp was created by pouring liquid precursor of UV-curable polymer (MD700) mixed with 1% photoinitiator (Darocur^®^ 1173) onto the Si master [32] following with flood lamp UV curing at room temperature (Dymax ECE 2000 UV, Torrington, CT, 2 min). The working stamp was used to hot emboss COC substrates (Zeonor 1060 R) under vacuum at 10 kN and 140 °C for 10 min. Demolding was done manually after the temperature of embossed stack dropped below 90 °C.

### 2.3. Details of Polymer Resins Used in The Membrane Fabrication

Liquid precursor of polymer MD700 was purchased from Solvay Specialty Polymers USA. MD700 is a perfluoropolyether (PFPE)-urethane dimethacrylate. UV curable MD700 resin was mixed by the precursor of polymer MD700 with 1% of photoinitiator Darocur^®^ 1173 (2-hydroxy-02-methyl-1-phenyl-propan-1-one, BASF Corporation, Vandalia, OH, USA). The viscosity of the UV curable MD700 is about 581 cP at 25 °C. The same UV curable MD700 polymer was used to fabricate the working stamps and membranes.

The medical adhesive UV resin 1161-M was purchased from DYMAX (Torrington, CT, USA). The viscosity of 1161-M is about 300 cP at 25 °C. The optical adhesive UV resin NOA84 was purchased from Norland Products Inc. (CRANBURY, NJ, USA). The viscosity of NOA84 is between 40–75 cP at 25 °C.

The UV curable EBECRYL 3708 resin was made from a mixture of Bisphenol-A epoxy diacrylate (50% in weight) and tripropylene glycol diacrylate (TPGDA). Both EBECRYL 3708 and TPGDA were purchased from Cytec (Allnex Canada Inc., Ottawa, ON, Canada). The viscosity of the UV curable EBECRYL 3708 resin is 604 cP at 23 °C. UV curable CUVR1534 resin was made from a mixture of UVACURE 1500 (from Allnex Canada Inc., Etobicoke, ON, Canada) and CAPA^TM^ 3035 (Perstorp Holding AB, Sweden), in a ratio of 50:50 by weight, mixed with 1% UVCURE 1600 photoinitiator. UVACURE 1500 is a class of cycloaliphatic di-epoxide resin, while CAPA^TM^ 3035 is a cyclic acylphosphoramidate polymer. The viscosity of CUVR1534 is 806 cP at 25 °C.

MD6945 thermoplastic elastomer polymer was made by dissolving pellets of Kraton^®^ MD6945M polymer (Kraton polymer llC, 15710 John F. Kennedy Blvd. Suite 300, Houston, TX, USA) in chlorobenzene with concentration of 14 wt. % (ACS reagent 99.5%, SIGMA-ALDRICH, St. Louis, MO, USA).

### 2.4. Polymer Filling by Capillary Force

In order to completely fill the PVA scaffolds with pre-polymer, two different strategies were employed. First, a cover coated with a thin layer of PVA was placed on top of the pillar template to make a closed PVA scaffold with a single open entry. A drop of pre-polymer was dispensed over this entry port and the sandwich was placed under vacuum for several minutes to evacuate the air trapped between the PVA template pillars [26]. Once the vacuum was vented, the polymer resin was pulled into the space between the pillars automatically filling the PVA scaffold. The advantage of this method is that there is no specific requirement for the polymer resin as long as it is not particularly volatile or viscous nor water soluble. The disadvantage is that it is not generally applicable to thermal setting polymer membranes because solvent evaporation from the resin through the single-entry port is typically a very slow process. Here, we focus on an alternative arrangement where the PVA scaffold is filled strictly by capillary-force-driven fluid displacement. This simplifies the overall process and can be readily scaled for mass production.

Casavant et al. [33] reported that liquids could propagate spontaneously by capillary force along micro-channels, even without floor and ceiling (suspended flow), if the internal pressure *P*, which is the variation of free energy per unit volume *dE*/*dV*, at the liquid front is negative.
(1)$$P=\frac{dE}{dV}=\gamma_{LG}\left(\frac{dA_{LG}}{dV}-\cos\theta \frac{dA_{SL}}{dV}\right),$$

Here, dA is the variation in area, with the indices *L*, *S*, and *G* representing liquid, solid and gas phases, respectively. The contact angle, *θ*, is between the fluid and the solid surface, and *γ_LG_* is the surface energy of the fluid between liquid and gas. With the reservoir pressure being set to zero and given that the flow condition is P<0, the condition for SCF can be written as:(2)$$\frac{dA_{LG}}{dA_{SL}}<\cos\theta ,$$

For an open system of PVA posts, it can be simplified as a number of long rectangular channels with a chain of semi-circle posts at each side of the channel with period *L* and radius of the circle *r* in parallel. Here, we just take an example of a microstructured PVA template with an array of micropillars of height *h*, radius *r* and pitch *L* (square configuration). The basic unit of this long channel consists of a box with dimensions *L* × *L* × *h* with four quarter pillars whose radius is *r* and a segment of open channel as shown in Figure 1. In this simplified system, $dA_{LG}=L^{2}-\pi r^{2}$ and $dA_{SL}=L^{2}-\pi r^{2}+2\pi rh$.

Equation (2) becomes:(3)$$\frac{L^{2}-\pi r^{2}}{L^{2}-\pi r^{2}+2\pi rh}<\cos\theta ,$$

Since the left side of Equation (3) is always smaller than 1, the above equation holds true as long as the contact angle is small enough. The cavity of a PVA scaffold can, therefore, be simply and automatically filled by SCF with polymeric resin.

To estimate the efficiency of the SCF filling in the PVA scaffold system, we assume an array period for the PVA posts of *L* = 60 µm, post radius of *r* = 10 µm and a post height of *h* = 80 µm_,_ which gives 0.3953 for the left side of Equation (3). The condition holds true as long as the contact angle of the resin with PVA surface is smaller than 66.7°. Since the surface of PVA is highly hydrophilic, most UV curable polymers will wet the pillar structures and the condition for SCF is satisfied. Table 1 shows the static contact angle of six UV curable polymer resins and one thermal set elastomer measured on the surface of unstructured PVA.

The contact angle of EBECRYL3708 is larger than 70°; in order to assist the SCF filling for this UV resin, the PVA surface was first treated with O_2_ plasma (100 W, 100 mTorr, 1 min), resulting in a static contact angle < 10° between the PVA surface and the resin.

The lateral filling velocity across the template surface area is directly related to the filling dynamics of the polymer resin into the micro-structured PVA surface. The roughness of a surface can enhance both the wetting (hydrophilic) and non-wetting (hydrophobic) nature of a liquid on a solid surface. When the Young’s contact angle on a flat/surface is less than 90°, roughness will reduce the apparent contact angle, leading to super-hydrophilic/super-wetting case. If the Young’s contact angle is larger than 90°, the roughness will increase the apparent contact angle, leading to super-hydrophobic/super-anti-wetting case. For a system with a micro-structured surface consisting of an array of micropillars with radius *r* and period *L,* with pillar density *ϕ_s_* = *πr*^2^/*L*^2^, the SCF of the liquid is possible via the menisci that forms locally at each pillar. This meniscus front enables the liquid to reach neighboring pillars. This wicking behavior is more accurately a hemi-wicking phenomenon, an intermediate state between spreading and imbibition [33,34]. The driving force for hemi-wicking is given by the following equation for conditions where the top surface of the pillars is dry:(4)$$F=\frac{dE}{dx}=\left(f-\emptyset_{s}\right)(\cos\theta -\cos\theta_{c}),\;$$
where *f* = 1 + 2*πrh*/*L*^2^,
(5)$$\cos\theta_{c}=\frac{1-\emptyset_{s}}{f-\emptyset_{s}}=\frac{L^{2}-\pi r^{2}}{L^{2}-\pi r^{2}+2\pi rh},$$
here the critical contact angle is *θ**_c_*, while *θ* is the Young’s contact angle.

The driving force depends on the surface energy of the liquid (polymer resin), the contact angle of the polymer resin on the solid surface and the surface microstructures. This capillary force is generally resisted by viscous forces due to the flow of the liquid (polymer). By balancing the capillary driving force with the viscous resistance, Washburn [35] deduced the time-dependent filling height in a material made of porous holes as follows:(6)$$l^{2}=tR_{eff}\gamma \cos\theta /2\eta ,$$
where $R_{eff}\;$ is the effective radius of the cavity, *γ* is the surface energy of the liquid and the *η* is the viscosity of the liquid. As an estimate for the above-mentioned system, where $R_{eff}$ is taken as 20 µm and a 30° contact angle for polymer MD 700 (viscosity *η* = 581 cP) [36]. The capillary progression length is 435 µm/s or 2.61 cm/min. In our open PVA micro-structured scaffold micro-fluidic system, the filling is 2D rather than 1D, so the filling depends on the properties of the polymer, as well as the microstructure of the device, including the period, pitch and diameter of the pillars. Generally, we found that the system filled rapidly by capillary action; an area of 16.5 mm by 33 mm filled in about 5 min.

The top surface of the pillars can be wetted during the progression of the polymer film, but this wetting is unstable. A droplet placed directly on top of the pillars will eventually penetrate into cavities, dewetting from the pillar tops—the typical Wenzel wetted state [37] as long as there is not excess polymer volume to otherwise affect the flood structure. To avoid overfilling, we controlled the dispensed volume used in the filling process. It was practical to create a wide groove surrounding the microstructured area to act as an overflow reservoir. This reservoir has a dual purpose to both speed up the filling process and act as an overflow reservoir for any excess polymer, ensuring open-through polymer membranes.

### 2.5. Fabrication of Open Through-Hole Polymer Membranes

UV polymer resin is dispensed into the groove surrounding the structured pillars on the intermediate PVA scaffold. The spaces between the pillars fill automatically with the resin. Once all the cavities in the PVA template are completely filled, the polymer resin is cured by UV exposure at room temperature for one minute with a flood lamp UV curing system (Dymax ECE 2000 UV, Torrington, CT, USA). The polymer filled PVA template is then submerged in water and dissolves with ultrasonic agitation for 5 min. Following separation, the as-cured polymer membrane is carefully dried with a reduced flow N_2_ gun.

The UV curing step is done uncovered and under ambient conditions. The surface of the UV resin is, therefore, exposed to air and does not fully cure due to oxygen inhibition common for most free radical UV resins. This could be solved by adding a drop of organic solvent on top of the resin to strip off the oxygen molecules absorbed on the surface after the initial curing step and re-exposing to UV light can fully cure the surface of the resin. Alternatively, full curing can be achieved in a controlled environment with O_2_ below 5 ppm in a Glovebox (UNILAB Pro., MBRAUN, Stratham, NH, USA) and a UV flood energy of 0.3 J/cm^2^ (SpectroLINKER^™^, XL-1000 UVC (254 nm), MBI Lab equipment, Farmingdale, NY, USA).

Although the PLA sacrificial templates used in the fabrication of small diameter polymeric membranes are biodegradable, the decomposition time of PLA in ambient conditions is in the order of months and years, and it is, therefore, not practical to separate the membrane from PLA templates in this manner. We used Nano-strip^®^ 2× (Amplex Chemical Products Ltd., Pointe-Claire, QC, Canada) to fully dissolve the PLA scaffold to make MD700 polymer membranes. MD700 is a bifunctional perfluoropolyether-urethane methacrylate and is resistant to most corrosive chemicals. After UV curing, the PLA template filled with MD700 polymer was immersed in Nano-strip^®^ 2× overnight. Once the PLA template was completely dissolved and the polymer membrane separated and rinsed multiple times in DI water, it was dried by a stream of N_2_ blow.

Likewise, Zeonor 1060R polymer is a class of thermoplastic cyclic olefin copolymer (COC) with good resistance to most chemicals, such as acids, bases and polar solvents, but poor against nonpolar solvents, such as hexane, toluene and other hydrocarbon-based oils. It is challenging to use Zeonor 1060R as a direct sacrificial substrate to make open-through-hole polymer membranes using the above methods because it is not readily dissolved without also attacking the polymer as well. Nevertheless, some polar solvents can swell, but not permanently damage the as-cured polymer membrane. In this case, instead of dissolving the sacrificial substrate in solvent, the swelling of the polymer membrane in specific solvents can in fact release the UV cured polymer from an easy to emboss Zeonor 1060R-based scaffold. The CUVR1534 polymer, for example, swells when immersed in methanol and can, therefore, be separated from a Zeonor 1060R template. After UV curing, the sample was placed in methanol (98%) for several hours, after which, the UV cured polymer membrane spontaneously released from the Zeonor 1060R microstructured scaffold, allowing the retrieval of a standalone CUVR1534 polymer through-hole membrane.

### 2.6. Details of Contact Angle Measurement, SEM Structural Characterization and E-Beam Evaporation

The static contact angles of polymer resins on the surface of 300 µm thick PVA thin films were measured by contact angle goniometer (Model 200, Ramé-hart Instrument Co., Succasunna, NJ, USA) and determined by DROPimage advanced (Version 2.0.10) software. Each reported measurement is an average of three independent experiments. Typical measurement error for these contact angles is ~2°.

The micro-nanostructured samples for both intermediate templates and polymer membranes were characterized by scanning electron microscopes (SEM, Hitachi, S-4800, Schaumburg, IL, USA). Sputtered 10 nm platinum thin films (LEICA EM ACE600, LEICA, Concord, ON, Canada) were coated on the SEM samples to avoid SEM charging effects during observation. For ease of handling, the SEM samples of a thin polymer membranes were prepared by adding a piece of Si substrate to the back side as a support. SEM images were taken both from top view and bird’s-eye view at a tilt angle of 75°.

For fabrication of an array of Gold (Au) and Palladium (Pd) nanodots, a section of polymer membrane with hierarchical micro-nano holes was placed on a Si substrate with nanoholes facing down. Au and Pd thin films were deposited using an e-beam evaporator (Kurt J. Lesker). The substrate was located about 300 mm above the e-beam gun. The angle of incidence of the metal vapor on the substrate was 90°. The base pressure of the chamber was kept below 5 × 10^−6^ mbar. The e-beam gun voltage and current were set to 10 kV and 60 mA for Au, and 40 mA for Pd, respectively. Both Au and Pd films were 30 nm thick and deposited at a rate of 1/s.

## 3. Results and Discussion

### 3.1. Large Area Through-Hole Polymer Membranes in Different Materials Using PVA Templates

Figure 2A shows an SEM image of a PVA template with an array of pillars with a diameter of 16 µm and height of 80 µm. Figure 2B shows an SEM image of the bottom side of a CUVR1534 polymer membrane whose fabricated thickness corresponds well to the height of the PVA pillars. Note that the CUVR1534 membrane shows high fidelity replication of the bottom “round” at the base of the PVA scaffold pillars on the membranes bottom surface. The cross section of the membrane pores clearly demonstrates that they are open through and straight, the diameter of the pores is approximately 16 µm. The top surface of the membrane is shown in Figure 2C. We observe a convex surface topology around each pore that is consistent with a meniscus shape. This indicates that the adhesive force between CUVR1534 and the side wall of the PVA pillars is larger than the cohesive energy of the resin. The local convex shape at each pore in the CUVR1534 membrane is retained after UV curing.

By using the PVA intermediate templates and applying the SCF resin filling method, we successfully fabricated a series of polymer membranes, including the free radical UV resin EBECRYL 3708, MD700, optical adhesive NOA84 with high refractive index and medical adhesive 1161-M. It was also found that PDMS can also spontaneously fill the PVA cavities, although the filling speed is not as fast as for the lower viscosity UV-cured resins. After the cavities of the PVA scaffold were fully filled with PDMS, the PDMS was cured in an oven (80 °C, 2 h). A PDMS through-hole membrane was then obtained by dissolving the PVA scaffold in deionized water. We attempted to make a MD 6945 thermoplastic elastomer membrane, but scaffold filling was successful only over small areas (few mm by few mm). Even though the contact angle of MD 6945 on PVA surfaces is small enough to drive filling, the chlorobenzene solvent in MD 6945 polymer evaporates too quickly, preventing the resin to propagate over a larger area. As shown in Equation (6), the resin filling speed increases with the capillary force, for example, the increasing of the surface energy and the decreasing of the static contact angle of the resin on the template. The resin filling speed increases with the decreasing of the viscosity of the resin too. We noticed that the filling speed of 1161-M and NOA84 resins was faster than for MD700, EBECRYL3708 and CUVR1534 resins. The filling speed can be increased by reducing resin viscosity through dilution with organic solvents. However, open array filling also requires low volatile solvents, limiting the use of solvents, such as chlorobenzene, used in the MD 6945 polymer resin to reduce viscosity.

Figure 2D shows an SEM image of the top side of a fabricated PDMS membrane. The membrane consists of two hierarchical layers. A square microhole (200 µm by 200 µm) was covered by a thin membrane layer with an array of 5 µm circular pores. The surface topology of the PDMS membrane around the holes is only slightly convex, instead it looks like it is quite flat, indicating that the adhesive energy between PDMS and the PVA pillars is comparable to the cohesive energy of PDMS. However, careful examination for the SEM image in Figure 2E for the 1161-M membrane shows, that the surface topology around the holes has a concave rather than a convex shape, which means that the adhesive energy between 1161-M resin and the PVA pillars is smaller than the cohesive energy of the 1161-M resin. Figure 2F is a SEM image of the top side of a MD6945 membrane. It is clearly observed that the holes are open through while the surface topology of the membrane around them has a convex shape, indicating that the adhesive energy between MD6945 and PVA pillars is larger than the cohesive energy of MD6945.

In order to test the ability to fabricate polymer membranes over large areas, we designed a photomask with a 90 × 90 mm^2^ patterned area that consists of four 44 × 44 mm^2^ patterned regions, each surrounded by a 2 mm wide channel used as groove reservoir for the resin filling. Figure 2G shows a photo of the PDMS mold fabricated based on this design. The dashed lines around the patterned area indicate the filling reservoirs in the PVA templates. In principle, one can get four pieces of membrane with areas of 44 × 44 mm^2^ after the completion of the process. Figure 2H shows a photo of one piece of CUVR1534 polymer membrane with a thickness of 30 µm and pore size of 15 µm; the pores are arranged in square configuration with a pitch size of 30 µm. We also successfully fabricated a membrane with 15 µm pores arranged in a hexagonal configuration, the distance between any two pores is fixed at 22 µm, for an overall 42% porosity for this specific membrane design.

### 3.2. Polymeric Membranes with High Aspect Ratio Pores

The aspect ratio (AR) of the membrane thickness to pore diameter is eventually limited by the AR of the PVA microposts used in the template, which is, in turn, determined by the silicon master mold and the mold replication process. The Si master mold is fabricated by DRIE; the AR in DRIE can easily go up to 30:1 [38]. The actual PVA intermediate scaffold is itself casting from a PDMS replica of the DRIE silicon master. The AR of the PVA pillars will determine the AR of the final membrane pores. The highest pore AR is, therefore, limited by the tallest achievable PVA pillar AR.

Figure 3A,B show the SEM images of the PVA templates used in the SCF process. The pillar diameter of the PVA templates shown in Figure 3A is 6.0 µm at the bottom and 5.8 µm at the top. This small reduction was caused by the over etching of the pillar tops (about one tenth of the pillar height) during the silicon DRIE process. The diameter of the pillars in the other PVA template, shown in Figure 3B, is 7.7 µm. The height of the pillars for both templates is approximately 100 µm. Therefore, the AR of height to diameter of the pillars is 16.7 and 13.0, respectively. This is in good agreement with the AR achieved in the membranes. To the best of our knowledge, these are the highest aspect ratios achieved so far for polymer membranes having straight, open-through holes. Previously, the aspect ratio of open-through-hole membrane made by a combination of phase separation micromolding and float-casting was on the order to 1 [22]. The AR of the open-though hole in polyurethane acrylate membrane reported by Cho was also around 1 [23] and an AR of ~3 was reported by Striemer in porous nanocrystalline silicon (pnc-Si) membrane using silicon fabrication techniques [7]. An AR of 12 was achieved in Poly(methyl methacrylate) membrane fabricated by nanoimprinting using metal molds [19]. Figure 3C,D are the SEM images of the bottom side of two NOA84 membranes fabricated from the PVA pillars shown in Figure 3A,B. The images shown in Figure 3E,F, clearly show open-through pores for both membranes. The surface topology on the bottom side of the membranes around the holes has a micro-concave shape. This is replicated directly from the original form of the pillar bases on the Si master, as a result of the quasi-isotropic DRIE etch process. Figure 3G,H show the top side of the two membranes. The cross-section view shows that the diameter of the pores on the top side of the membrane is slightly smaller than at the bottom. This is especially obvious as the diameter of the pillar structures was reduced by over etching the pillar tops during DRIE. Careful examination of the surface topology of the membrane around each local pore opening again shows the micro-convex shape, resulting from the stronger adhesive energy between NOA84 and PVA pillar, relative to the NOA84’s cohesive energy. The adhesive energy between the resin and the substrate is due to the polar forces or direct bonds that can form between reactive sites in the resin and reactive or polar sites on the surface of a substrate. Some polymer resins have pendant hydroxyl groups along their chains, which can form chemical bonds or strong polar attractions to oxide or hydroxyl surfaces on a substrate. Therefore, the adhesive energy between a resin and a substrate can be fine tuned by adding or removing some chemical components in the resin. Although the micro-convex shape of the pore opening is not desirable for an application when the membrane is required to have an air-tight bond on a substrate, the shape of the pore opening has little impact on the properties of the membranes needed in most other applications, such as for integrated filters in microfluidic devices, used for sample preparation in clinical, food or environmental safety.

### 3.3. Polymeric Membranes Using PLA Templates

Figure 4A–C show a picture of a silicon master mold, a photo of an intermediated FEP working mold and a photo of a PLA substrate on an 8’’ diameter silicon wafer as-replicated from the master Si mold. Since there are no filling reservoirs built into this design with 2 µm diameter micropillars, the entire 4.8 cm × 4.8 cm PLA template area was filled by dispensing a drop (12 µL) of fluorinated UV curable MD700 resin at the center of the PLA template. After the open PLA micropillars structures (hexagonal close packed arrangement) were fully filled with the UV resin, the template was UV cured with flood energy of 0.3 J/cm^2^ inside a controlled environment glovebox with O_2_ levels below 5 ppm to avoid the oxygen inhibition. Figure 4D shows a picture of a fabricated MD700 membrane with a size of 4.8 cm × 4.8 cm. The area marked by a red circle was the region, where the drop of MD700 UV resin was dispensed by pipetting. It was found that the membrane pores in that area were not fully open; here, the pillars were covered by the MD700 UV resin as volume of resin was slightly more than the total volume of the cavities in the PLA template.

The SEM images of the membrane and the PLA template are shown in Figure 4I,J. The SEM images of a Si master mold are shown in Figure 4E,F and the SEM images of the PLA template with micropillars replicated from the silicon master are shown in Figure 4G,H. These indicate that the PLA micropillars were well replicated from the silicon master mold, the diameter of the pillars is in the order of 1.8 µm with a height of 8.8 µm, both are in good agreement with the observed pore sizes and thickness of the as-fabricated MD700 membrane shown in Figure 4I,J. Here, the overall porosity of the membrane is 43.3%.

### 3.4. Polymer Membranes with Hierarchical Micro-Nano Apertures

Figure 5A,B show SEM images of a hot-embossed Zeonor 1060R substrate, indicating that the complex pillars are well-replicated from the original silicon master mold. On top of each 15 µm diameter, 30 µm high micropillar there are an array of 600 nm high, square nanopillars, with dimensions of 220 × 240 nm. Note the nanopillar size is smaller than the target 300 nm value due to slight over-etching during the fabrication process and the original square shape being rounded during processing. There are some defects in the nanopillar array, Figure 5C,D, particularly close to the edge of the micropillars, likely due to incomplete filling or damaged incurred during the various replication, hot embossing and demolding steps. The microstructures that appear around the 15 µm pillars, as well as the additional needle structures on the floor between the micropillars are attributed to passivation effects during the DRIE process. The mask used in the DRIE etching was patterned by photolithography on the positive photoresist AZ5214E. The full area array of Si nanopillars were patterned across a 10 × 10 mm^2^ area. The nanopillars that were protected by AZ5214E photoresist, creating the micropillars, were protected during the DRIE process and remain intact on the top of the micropillars. It is expected that most of the unprotected Si nanopillars are destroyed during the 30 µm deep etch; however, we do see some silicon pins with sizes ranging from 200 to 500 nm with height up to 5 µm randomly distributed on the bottom of the wafer in Figure 5A,C due to micromasking effects from the nanopillars during the DRIE process. Similarly, there are DRIE etching artifacts, such as the nanostructured needles, surrounding the perimeter of the micropillars themselves.

In order to test the feasibility to fabricate polymer membranes with sub-micrometer open though holes by using the SCF filling method, cationic CUVR1534 resin was selected as the starting material for the membrane fabrication, and hot-embossed Zeonor 1060R, with a combination of nanopillars on top of micropillars was selected as a template. The static contact angle of CUVR1534 on the plane surface of a Zeonor 1060R substrate after hot embossing (without microstructures) is 33°. This is smaller than the critical contact angle required for SCF filling of an array of 15 µm diameter, 30 µm tall micropillars with a pitch size of 30 µm ($\theta_{c}=70.2^{{}^{\circ}})$. Therefore, the CUVR1534 resin easily fills the space between the Zeonor 1060R template micropillars. The critical contact angle required for SCF filling the array of square nanopillars (300 × 300 nm^2^ on a centered hexagonal arrangement) on top of the micropillars is $\theta_{c}=76.4^{{}^{\circ}}$ It becomes 71.4° after correction using the size of the actual nanopillars (i.e., 230 nm per side in average). The CUVR1534 resin can, therefore, easily wet the top of each micropillar decorated with an array of nanopillars. The size variation of the nanopillars does affect the critical contact angle, but the working window is quite wide, as long as the height of the pillars is around 600 nm. Figure 6 shows the plot of the critical contact angle vs. pillar size for three different heights (for pillars arranged in centered hexagonal configuration and with nearest neighbor distance fixed at 600 nm). When the height of the pillars is fixed at 600 nm, the SCF condition holds true for the array of nanopillars, even when the pillar diameter drops below 50 nm. From a fabrication point of view, however, it is quite challenging to produce 50 nm diameter, 600 nm tall Zeonor 1060R nanopillars and an aspect ratio of 12. As the height of the nanopillars decreases, the critical contact angle becomes smaller too, which makes SCF filling more difficult. When the height of the pillars decreases below 100 nm, the SCF condition breaks down for nanopillars with a diameter smaller than 150 nm because the critical contact angle is approaching the static contact angle of the resin on the substrate, as shown by the dotted red line in Figure 6. Although the static contact angle is still smaller than the critical contact angle when the pillars are bigger than 150 nm, it might not be practical to wet the surface along the top of the micropillars. If we take into consideration the degree of difficulty of fabricating a device with such complex pillar structures, both successful fabrication and SCF filling are limited by the aspect ratio of the nanopillars. For example, selecting a reasonable AR of 4, the region delimited by the dotted red and blue lines represents the working window of SCF filling.

Figure 5E shows a picture of a CUVR1534 membrane after carefully drying with N_2_ against a backing FEP foil. The dimensions of the CUVR1534 membrane are 44 × 44 mm^2^, which is the same size as the membrane that we made by using the PVA sacrificial template. Figure 5F shows an optical microscopy image of the membrane at the marked region where the microholes are open at the bottom side but closed with a very thin membrane with nanoholes on top side, confirmed by the SEM images shown in Figure 5G,H.

The details of the open-through nanoholes are summarized in Figure 5G–P. In Figure 5G, the surface of the bottom side of the membrane looks somewhat rough or porous. This resulted from the presence of some micro-structured needles on the fabricated Zeonor 1060R template. They are not open through because the height of those needles is less than 5 µm, as shown in Figure 5A,C. From the cross-section view of one of the microholes, it is found that the micrometer scale featured on the wall of the pore were well-replicated from the features found on the Zeonor 1060R micropillars template, shown in Figure 5A,C. The microhole is covered by a very thin layer on the other end, and as imaged under a high magnification view, as shown in the insert of Figure 5G, the thickness of this thin membrane layer was found to be approximately 550 nm, which is consistent with the height of the nano pillars shown in Figure 5B, and these holes are seen to be open through. Figure 5H is a SEM image of the membrane viewed at a tilt angle (75°), the top side of the membrane is facing up. From this cross-section view, the 15 µm holes are closed by a thin layer membrane on top while they are opened at the bottom side. Under high magnification, again, it is found that the micrometer holes are open through. Figure 5I,J, are elevated view SEM images of the membranes from above for the bottom and top side, respectively. Figure 5I,J were taken under the same magnification. Although the image gives the impression that the diameter of the microholes at the bottom is significantly larger than the top, careful inspection under higher magnification (Figure 5K) shows the diameter of the microhole at the top side should be “bigger” than it appears in Figure 5L, as it includes the ring marked by dashed red lines. The contrast of this ring is darker than that of the area enclosed by the small dashed red line circle because the thinner layer in this area is not open through, which is consistent with the SEM image of the pillars shown in Figure 5A,B. After this correction, the diameter of the microhole at the top side is about 15.4 µm, which is slightly smaller than that (16 µm) at the bottom side (Figure 5K). There is also a smaller difference in the dimensions of the micrometer holes between the bottom and top sides. The higher magnification SEM images, shown in Figure 5M,N, indicate that the diameter of the open through hole at the bottom side is around 250 nm (rounded square) while it is about 200 nm to 210 nm at the top. Some size variations at the top side were also observed; pores are slightly smaller around the edge than in the central region. This is likely due to the non-uniformity of the nanopillars replicated from the Si master (variation in height, top of a pillar becoming more rounded and smaller, etc.). As shown in Figure 5O,P, an open-through hole at the edge of a microhole on the top side of the membrane as small as 70 nm was observed.

### 3.5. Application of Polymeric Nanoaperatures as a Shadow Mask

As one straightforward application example of these polymeric membranes, the nanoapertures were used as a stencil for the fabrication of metallic nanodots without the need complex nano-scale metal lift off techniques using multi-level, undercut resists. The top side of a membrane was put in direct contact with a Si surface (it could be conformably placed onto other plastic substrates as well). Following the deposition of metallic films by e-beam evaporation and removal of the membrane stencil, arrays of Au and Pd nanodots remained on the silicon wafer, as shown in Figure 7. The Au nanostructures could be used as surface-enhanced Raman scattering (SERS) or surface plasmon resonance (SPR) substrates for biosensing applications [39,40].

## 4. Conclusions

We demonstrated a simple, yet robust, templating method to make freestanding polymeric membranes from an intermediate scaffold. A filling strategy based on spontaneous capillary flow into the microstructured template results in high-fidelity open through-hole membranes with regular pore sizes and densities for both UV-cured and thermoset polymers. Polymerisation was done by either UV or thermal curing, and the method was successfully applied to fabricate a series of flexible membranes across a broad range of materials, including cationic and free radical UV resins, such as CUVR1534, EBECRYL3708, MD700, optical adhesive with high refractive index NOA84, medical adhesive 1161-M and PDMS.

By applying the concept of geometric reinforcement via multiple size scales and multilevel architectures, we also demonstrated the fabrication of polymer membranes with hexagonal arrays of 200 nm diameter and 600 nm pitch nano-apertures and by using this polymer membrane as a stencil, we fabricated an array of Au nanodots and Pd nanodots as small as 130 nm on Si substrates.

The advantages of the present method enable significant improvements to the fabrication of polymer membranes with regular and open-through pores, including high aspect ratios (16.7), high porosity (42%) and very large surface area (44 × 44 mm^2^). The polymer membranes were also fabricated in a variety of different materials and for a broad range of pore sizes (from 100 µm down to 200 nm). These polymer membranes will find numerous applications across multiple fields of study, including microfluidic devices for sample preparation in clinical, food, or environmental safety, 3D microfluidic devices for general filtration, in pharmaceutical research and for novel drug delivery systems.

## Figures and Tables

**Figure 1 polymers-14-04228-f001:**
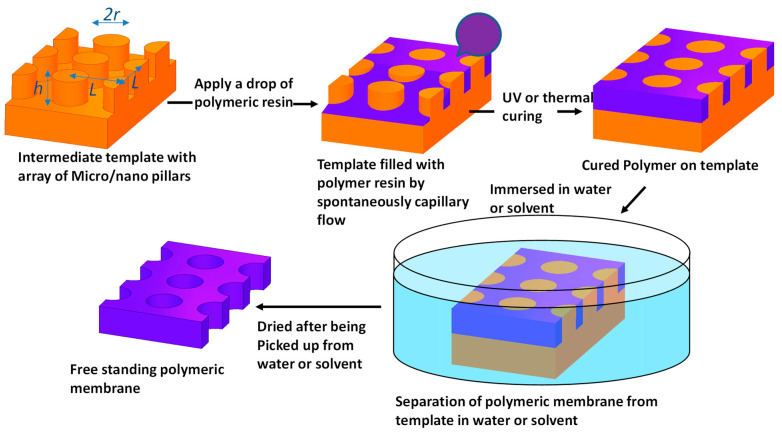
Process flow chart to make polymeric membranes via an intermediate template and SCF filling. The polymeric membrane is separated from the template by dissolving the template in deionized water (PVA template) or by swelling-based lift-off in solvent-assisted separation (Zeonor 1060R template).

**Figure 2 polymers-14-04228-f002:**
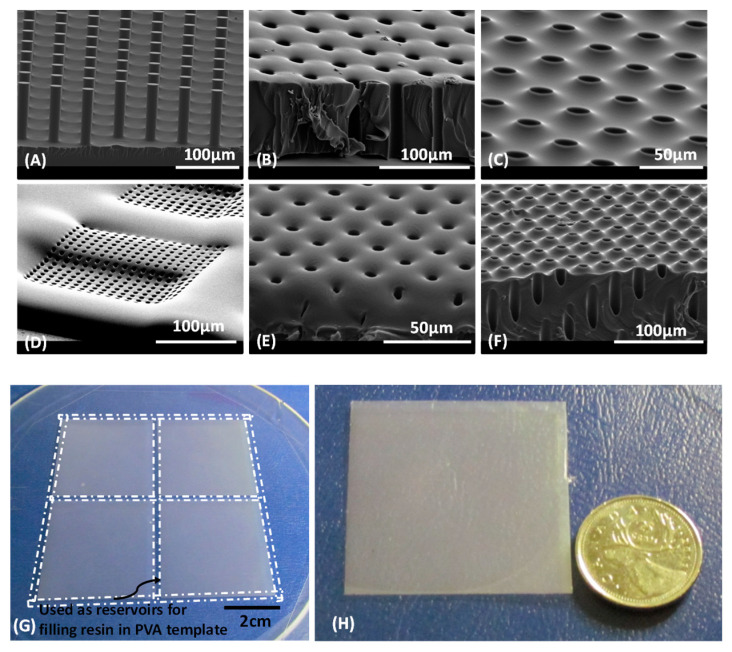
SEM images (**A**–**C**) of as-fabricated CUVR1534 membranes (80 µm thick, 16 mm by 33 mm area). (**D**) SEM image of the top side of a PDMS membrane. (**E**) SEM image of the top side of an 1161-M polymer membrane. (**F**) SEM image of an MD6945 membrane. (**G**) Wide area PDMS replica used in fabrication of PVA template. (**H**) One of four sections of a CUVR1534 membrane (30 µm thick, 40 × 40 mm^2^ area).

**Figure 3 polymers-14-04228-f003:**
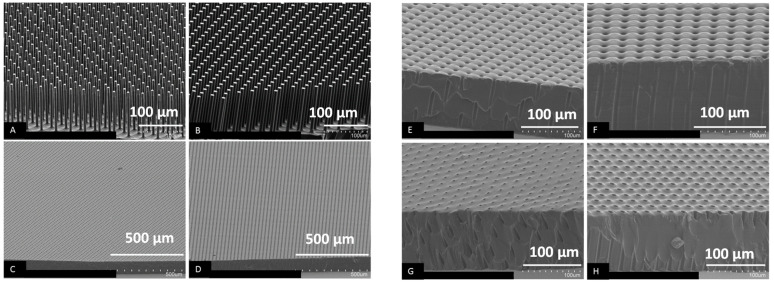
(**A**,**B**) SEM images of PVA pillars used for fabrication of polymer membranes. The diameter of the tip of a PVA pillar in (**A**) is 5.8 µm at the top and 6.0 µm at the bottom. The diameter of a PVA pillar in (**B**) is 7.7 µm. (**C**,**E**,**G**) and (**D**,**F**,**H**) are SEM images of the fabricated NOA84 membranes corresponding to the PVA pillars shown in (**A**,**B**), respectively. (**C**–**F**) are SEM images of the bottom side of these membranes, while (**G**,**H**) show top side.

**Figure 4 polymers-14-04228-f004:**
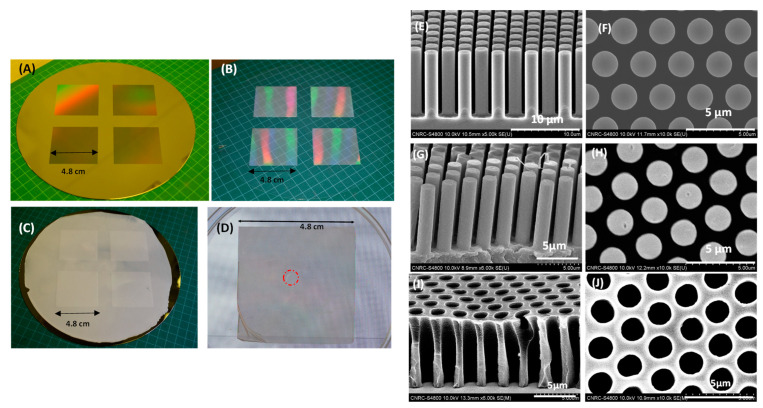
(**A**) Si master mold; (**B**) Intermediate FEP mold replicated from the Si master mold shown in (**A**); (**C**) Photo of a PLA template mold replicated from the FEP mold using HE; (**D**) A section of MD700 membrane (4.8 cm by 4.8 cm) area marked by a dotted red circle (defects) where the holes of the MD700 membrane are not open through. (**E**) SEM image of a Si master mold (tilt view); (**F**) SEM image of a Si master mold (top view); (**G**) SEM image of a sample cut from the PLA template mold (tilt view); (**H**) SEM image of a sample cut from the PLA template mold (top view); (**I**) SEM image of a piece of MD700 membrane replicated from the PLA template mold (tilt view); (**J**) SEM image of a piece of WS membrane replicated from the PLA template mold (top view).

**Figure 5 polymers-14-04228-f005:**
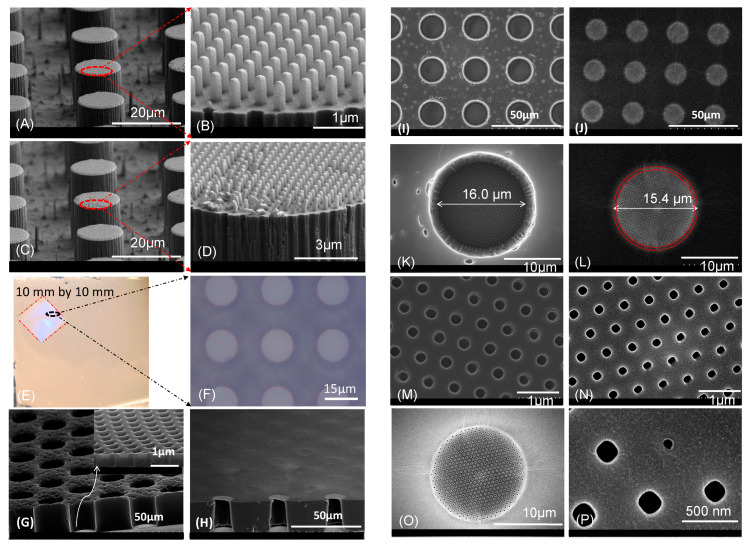
(**A**,**C**) SEM images of hot-embossed Zeonor 1060R substrates with a combined hierarchical nanopillars and micropillars. Some nanopillars defects, resulting from the demolding process, can be seen. (**B**,**D**) Close-up SEM images of the pillar tops shown in (**A**,**C**). (**E**) A piece of CUVR1534 polymer membrane (44 × 44 mm^2^), the region marked by dashed red lines is the membrane with hierarchical micro- and nano-holes. (**F**) An optical microscope image of the membrane inside the marked region. (**G**,**H**) SEM images (tilt at 75°) of a CUVR1534 membrane with combination of nano-/micro-open-through holes (shown from bottom and top sides, respectively). Insert in (**G**) is the cross-section view of the membrane with nano-holes. (**I**,**J**) are elevated view SEM images of the membrane from top for both the bottom and top side of the membrane, respectively. (**K**–**N**) are the SEM images of (**I**,**J**) at higher magnifications, the diameter of the open through hole at the bottom side is around 250 nm (rounded square), while it is about 200 nm to 210 nm at the top. (**O**,**P**) are the elevated view SEM images of the membrane at a location where an open-through hole of about 70 nm was observed at the edge of a 15 µm micro-hole.

**Figure 6 polymers-14-04228-f006:**
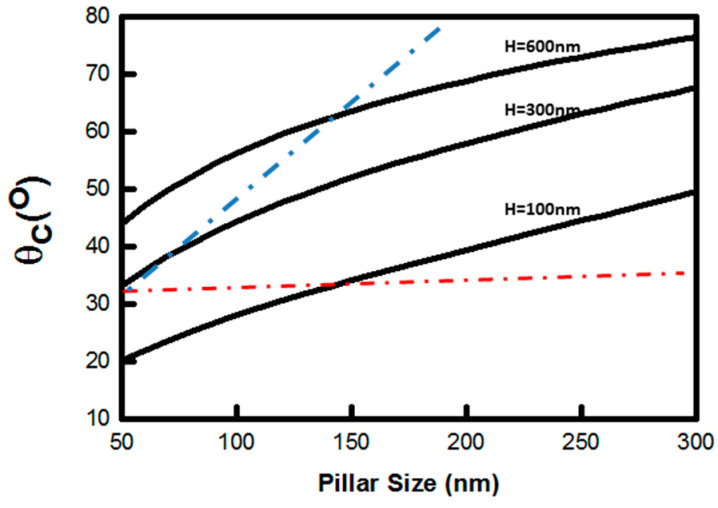
Plot of critical contact angle (*θ**_c_*) vs. pillar size for three different pillar heights. The dotted red line shows the value of the static contact angle of CUVR1534 resin measured on plane surface of hot-embossed Zeonor 1060R substrate (without microstructures). The dotted blue line was obtained after connecting the points where the maximum aspect ratio of the nanopillars is 4 for each *θ_c_* curve at different height of the pillar. The region enclosed by the dotted red and blue lines represents the working window of SCF filling of CUVR1534 resin on nanostructured Zeonor 1060R template.

**Figure 7 polymers-14-04228-f007:**
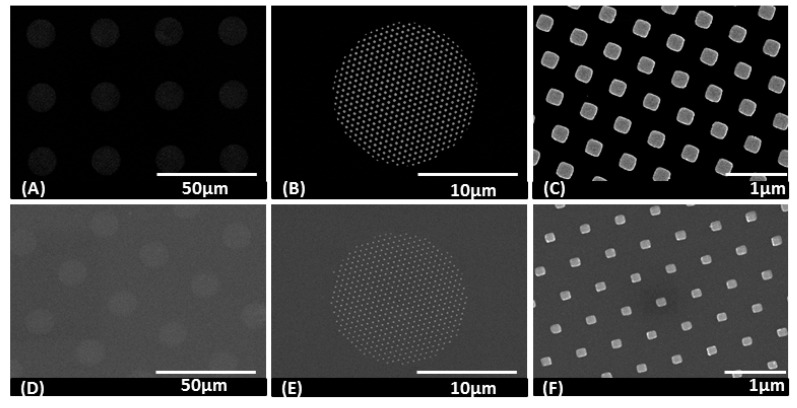
(**A**–**C**) SEM images of fabricated Au nanodots on a pieces of Si substrate using a polymer membrane with a combination of micro- and nano-holes as a stencil at different magnifications. The size of the Au dots is around 250 × 250 nm^2^ and 30 nm in height. (**D**–**F**) SEM images of fabricated Pd nanodots on a piece of Si substrate using a polymer membrane with a combination of micro- and nano-holes as a stencil at three different magnifications. The size of the Pd is about 130 × 150 nm^2^, with a height of 30 nm.

**Table 1 polymers-14-04228-t001:** Table of static contact angle of polymeric resins on the surface of PVA.

Material	CUVR1534	EBECRYL3708	PDMS	MD700	1161-M	NOA84	MD6945
Contact angle (°)	49.3	71.4	43.0	30.0	5.0	5.0	41.0

## Data Availability

Not applicable.

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
