# Peer review of "Facile Fabrication of Flexible Polymeric Membranes with Micro and Nano Apertures over Large Areas"

_polymers, 2022, doi:10.3390/polym14194228_

Round 1

Reviewer 1 Report

Dear Author,

The manuscript titled "Facile Fabrication of Flexible Polymeric Membranes with Micro and 2 Nano Apertures over Large Areas" is an interesting read. Here are my comments:

(1) Figure 2.1 font size needs to be increased as the text is very small for reading.

(2) Does the membrane shrinks after removal from the template solvent and after drying?

(3) Is there any effect on resolution or aspect ratio when moving from micro to nanostructures?

Overall, the work is really good to be accepted with minor revisions.

Author Response

Dear Reviewer 1,

Thank you very much for your careful evaluation of our manuscript and your valuable comments.  We have revised our manuscript accordingly, and would like to respond to your comments below.  

  • Figure 2.1 font size needs to be increased as the text is very small for reading.

To our understanding, it refers to both Fig.1 and Fig.2, the font size in Figure 1 & 2 is increased for reading.

  • Does the membrane shrink(s) after removal from template solvent and after drying.

There are almost no effects after removal from temperate and drying when water is used as solvent for dissolving the PVA template in the fabrication of a series of polymer membranes including CUVR1534, NOA84, MD700, EBECRLY3708, 1116-M, PDMS, and Nanostrip is used for fabrication of MD700. However, in the fabrication of CUVR1534 membrane with hierarchical micro-nano open-through holes using the Zeonor 1060R templates, CUVR1534 membrane swells in methanol, it was noticed that the holes were slightly bigger than the diameter of the pillars when it was immersed in the methanol, the membrane shrunk after removing from the methanol and drying. However, after drying, the pore size of the membrane is in good agreement with the size of the pillars of the template, indicating that it is finally determined by the size of the pillars of the template used for the membrane fabrication.

  • Is there any effect on resolution and aspect ratio when moving from micro to nanostructures?

In principle, there is no effect on resolution and aspect ratio when moving from micro to nanostructures because the method reported in this paper for fabrication of polymer membranes does not involve any mechanical peeling forces in the separation of membranes from the templates. It has advantages over many other methods to fabricate polymer membranes with higher aspect ratio. However, the aspect ratio and porosity of polymer membrane with straight open through holes are determined by the aspect ratio and density of the pillars of the template used in the fabrication. It is getting challenge to fabricate the intermediate templates with higher aspect ratios when moving from micro pillars to nanopillars especially when the density of the pillars is increased. Actually the resolution and aspect ratio become worse when moving from micro to nanostructures. But it could be improved by exploring advanced techniques in the fabrication of intermediate templates.   

  We hope that you find these changes appropriate and we look forward to hearing from you. 

Sincerely yours,

Kebin Li

Reviewer 2 Report

This manuscript can be accepted for publication after major revisions, see the followings:
*English should be improved.
*The Abstract should be improved.
*Better description and explanation on the figures (Figures 4 to 7).

 * The novelty is not clear.
*The Conclusion could, also, be improved.

Author Response

Dear Reviewer 2,

Thank you very much for your careful evaluation of our manuscript and your valuable comments.  We have revised our manuscript accordingly, and would like to respond to your comments below.  

  • English should be improved.

The revised manuscript (please see the revised manuscript with track changes) has been extensively checked and edited by one of co-authors who is a native English speaker. We believe that the English of the revised manuscript is improved. 

  • The abstract should be improved.

We have revised and improved the abstract accordingly. Please see the revised manuscript with track changes.

  • Better description and explanations on the figures (Figure 4 to 7).

After following the valuable suggestion point (1) from reviewer 3, the experimental methods and results & discussions are separated in this revised manuscript. Both the figure captions and descriptions in the text for each figure are revised, Figure 1 and Figure 5 are updated, we believe that the description and explanations on figures are improved. Please see the revised manuscript with track changes.

  • The novelty is not clear.

We believe that the novelty of our manuscript is in the fabrication of the polymer membrane with high performance, with high aspect ratio, high porosity and in large surface area. The novelty is in two aspects, namely the spontaneous capillary filling of polymer resin on micro-nanostructured templates as well as solvent removal of sacrificial scaffolds for separating the UV cured or thermal cured polymer membrane from the templated without undergoing any mechanical force. It has many advantages over other methods reported in the literature.

  • The conclusion could also be improved.

We have revised the conclusion and believe that it is improved. Please see the revised manuscript with track changes.

 We hope that you find these changes appropriate and we look forward to hearing from you. 

Sincerely yours,

Kebin Li

Reviewer 3 Report

Overall comments: In this work, the authors demonstrated the fabrication of open through-hole polymeric micro and nanostructured membranes over large areas (>16 cm2) via solvent removal of sacrificial scaffolds. Multiple polymer types were fabricated with the demonstrated method through a rapid UV curing process via cationic or free radical UV polymerization. Various pore sizes (200 nm - 100 μm) and membrane thickness (10 - 100 μm) were achieved. High aspect ratios and large areas were also successfully achieved along with membranes of different thicknesses and pore sizes. However, more descriptions and explanations of the experimental details and results are needed to improve the quality of the manuscripts. The manuscript could be recommended for publishing in Polymers after minor revision.

1. The authors do not have separate sections of experimental methods and results and discussion. Multiple experimental and instrumental details are missing, including: (1) details of polymer resin (chemistry, viscosity) (2) fabrication of the Si master mold for making the PVA intermediate template (3) contact angle measurement; (4) SEM measurement details (5) e-beam evaporation. Please check the experimental details and add accordingly

2. line 267: The abbreviation of TPGDA needs to be expanded.

3. line 270 – 283: How does polymer resin properties (concentration of the solution, type of the organic solvent, etc) affect the viscosity of the polymer solution and the filling speed? What types of polymer resin were considered to be evaporating too fast? A more detailed discussion is needed to guide the reader on the application of this method for different polymer resins.

4. Figure 3: Please add the diameters of pillars for the (A) and (B) PVA templates to the figure 3 legend to distinguish the figures more clearly.

5. line 323: How does the aspect ratio compare to other papers in the literature? More discussion on the aspect ratio is needed to help readers understand the impact of this fabrication method

6. line 333 – 336: What causes the differences in adhesive energy between the polymer resin and PVA pillar? Is there any impact of the shape of the pore opening on the properties of the resulting membranes? More discussion on this topic is needed. 

Author Response

Dear Reviewer 3,

Thank you very much for your careful evaluation of our manuscript and your valuable comments too.  We have revised our manuscript accordingly, and would like to respond to your comments below.   

  • The authors do not have separate sections of experimental methods and results and discussion. Multiple experimental and instrumental details are missing, including: (1) details of polymer resin (chemistry, viscosity) (2) fabrication of the Si master mold for making the PVA intermediate template (3) contact angle measurement (4) SEM measurement details (5) e-beam evaporation. Please check the experimental details and add accordingly.

The experimental methods and results and discussion are separated now. Details of polymer resin (section 2.3), fabrication of the Si master mold (section 2.2), contact angle measurement, SEM measurement and e-beam evaporation (section 2.6) are added in the experimental section. The description about the experimental methods involved in the results and discussion are removed accordingly. Please see the revised manuscript with track changes for details.

  • Line 267: The abbreviation of TPGDA needs to be expanded.

It is expanded in the section of 2.3 about the details of polymer resins.

  • Line 270-283: How does polymer resin properties (concentration of the solution, type of the organic solvent, etc.) affect the viscosity of the polymer solution and the filling speed? What types of polymer resin were considered to be evaporating too fast? A more detailed discussion is needed to guide the reader on the application of this method for different polymer resins.

The resin filling speed increases with the capillary force, for example, increasing of the surface energy and decreasing of the static contact angle of the resin on the template. The resin filling speed increases with the decreasing of the viscosity of the resin too. We noticed that the filling speed of 1161-M and NOA84 resins was faster than for MD700, EBECRYL3708 and CUVR1534 resins. The filling speed can be increased by reducing the viscosity of the resin diluted with much more organic solvent. But it requires that the solvent should be evaporating as slow as possible, otherwise, like the chlorobenzene used in the MD 6945 polymer resin doesn’t help to increase the filling speed. The discussion was added in the section of 3.1.

  • Figure 3: Please add the diameters of pillars for the (A) and (B) PVA templates to the figure 3 legend to distinguish the figures more clearly.

       The diameters of pillars are added in the figure caption. 

  • Line 323: How does the aspect ratio compare to other papers in the literature? More discussion on the aspect ratio is needed to help readers understand the impact of this fabrication method.

It is compared to other papers in the literature and more discussion on the aspect ratio is added. Please see the text added in the second paragraph in the section of 3.2.

  • Line 333-336: What causes the differences in adhesive energy between the polymer resin and PVA pillar? Is there any impact of the shape of the pore opening on the properties of the resulting membranes? More discussion on this topic is needed.

The adhesive energy between resin and a substrate is essentially due to the attraction forces which are usually polar forces or direct bonds that can form between reactive sites in the resin and reactive or polar sites on the surface of a substrate. Some polymer resins have pendant hydroxyl groups along their chains which can form chemical bonds or strong polar attractions to oxide or hydroxyl surfaces of a substrate. Therefore the adhesive energy between a resin and a substrate can be fine tuned by adding or removing some chemical components in the resin. Although the micro convex shape of the pore opening is not desirable for an application when the membrane is required to have an air tight bond on a substrate, the shape of the pore opening has little impact on the properties of the membranes used in most of other applications such as filters integrated in microfluidic devices for sample preparation in clinical, food, or environmental safety. Please see the last paragraph of the section 3.2 of the revised manuscript.

 We hope that you find these changes appropriate and we look forward to hearing from you. 

Sincerely yours,

Kebin Li

Round 2

Reviewer 2 Report

This article can be accepted.